# Cone-Beam Computed Tomography (CBTC) Applied to the Study of Root Morphological Characteristics of Deciduous Teeth: An In Vitro Study

**DOI:** 10.3390/ijerph19159162

**Published:** 2022-07-27

**Authors:** Jesús Ticona-Flores, Montserrat Diéguez-Pérez

**Affiliations:** 1Faculty of Biomedicine and Health Sciences, European University of Madrid, C. Tajo s/n, 28670 Villaviciosa de Odón, Madrid, Spain; jesus.ticona.f@upch.pe; 2Preclinical Dentistry Department, Faculty of Biomedicine and Health Sciences, European University of Madrid, C. Tajo s/n, 28670 Villaviciosa de Odón, Madrid, Spain

**Keywords:** primary teeth, root canal, pulp canal anatomy, cone-beam computed tomography

## Abstract

Pulp therapy in primary teeth is a challenge for a dentist, therefore, a better understanding of the anatomical characteristics of this tissue is essential to remedy these deficiencies. The aim of this study was to determine the morphological peculiarities of the root canals of extracted deciduous molars by Cone-Beam computed tomography (CBCT). As such, healthy molars without physiological resorption were collected and sanitized. After analyzing descriptive and inferential statistics, the results show that 56% of lower molar roots have a Weine’s type III canal configuration; the upper and lower second molars are significantly larger compared to the first, with a mean of 8.318 (±1.313) mm and 7.757 (±1.286) mm, respectively. Additionally, the palatine canals exhibited greater volume than the others, with a mean of 3.687 mm^3^. There are multiple discrepancies in the molars that have been studied in other investigations. The presence of a supernumerary root in the mandibular molars, a smaller dimension of the root canals and a more evident apical angulation are more obvious. The discovery of great anatomical versatility in the posterior dental group is an aspect applicable to root therapy.

## 1. Introduction

The preservation of the deciduous teeth until their physiological replacement is essential for chewing, the eruptive guidance of permanent teeth emergence and provides a stimulus for the correct development of the maxillaries. This development can be truncated by infectious pulp pathologies of bacterial or traumatic origin. In an attempt to prevent the premature loss, pulpectomy allows for an increase in the permanence of the primary teeth in the maxillaries [1,2].

Deciduous root canal therapy is a challenge for the professional. This is a delicate and meticulous procedure determined by the great variability in the morphological characteristics of the deciduous tooth root, the lack of knowledge of these and the clinical time constraints determined by the behavioral management of the child patient [2,3,4,5]. The exhaustive knowledge of pulp anatomy, especially root canal, increases the prognosis and success of pulp therapies, because the primary teeth have a highly complex root canal system. The capricious and variable anatomy, which has numerous accessory, reticular and recurrent canals, and apical delta ramifications, is not clinically appreciable with radiographic techniques, because the superimposition of the mineralized structures makes it impossible to observe faithfully in the buccolingual/palatal direction [2,6,7,8,9,10,11,12,13,14].

The complexity of the root canal systems affects the removal of the pulp tissue and favors the retention of organic residues, microorganisms and toxins. This could compromise the success of the treatment [6,11].

Certain in vitro study techniques, such as whitening by decalcification or methylene blue staining, alter the structure and penetrate the dentinal tubules, making it difficult to visualize and study root canals [11,15]. The CBCT imaging technique has helped complement conventional radiography, achieving a more complete evaluation of the morphology of root canal variations [8,13,16,17,18]. Micro-computerized tomography (micro-CT) is another technique, which can generate better three-dimensional reconstructions, reflecting more accurately and didactically the micrometric characteristics of the root canal [2,6,19,20]. However, it has several drawbacks, such as it is not suitable for clinical use and the studies based on it are still insufficient due to the risk it implies for the pediatric dentistry patient [6,8].

The interest of, and justification for, this research are based on the following. First, anatomical variability in different ethnic groups [3]. The scarce scientific literature describes these anatomical peculiarities, such as angulation, diameter and length of the root canal. Secondly, the need to understand the root canal anatomy, due to the high demand for pulpectomies and the increase in their success in terms of their prognosis [5,12]. In this context, we hypothesized the existence of undescribed root anatomical details, and were able to contribute novelties in this regard with our results.

The aim was to study the morphological peculiarities of the root canals of extracted deciduous molars by cone-beam computed tomography (CBCT).

## 2. Materials and Methods

This was an in vitro observational study, which was approved by the Ethics Committee of the San Carlos Clinical Hospital in Madrid (Spain), with internal code 21/375-E.

### 2.1. Sample Selection

The deciduous molars were collected, regardless of gender, age, or race of origin and extracted in different public and private dental offices. The inclusion criteria were maxillary and mandibular teeth without root resorption (without loss of dental hard tissue resulting from the physiological process of clastic cell activity) or at least one complete root; and extracted by prescription of orthodontic treatment (orientation of the eruption, dental crowding, etc.), or by exceeding its normal exfoliation time. In addition, caregivers who agreed to donate their child’s tooth

The exclusion criteria were teeth with a history of pulp treatment (pulpectomy or pulpotomy); with advanced carious or pulpal pathology (external, internal, perforating root resorption, or root canal obliteration); and anterior teeth (Figure 1).

For the sample calculation, a bilateral hypothesis was established, the power of 80%, the significance of *p* < 0.05; the research of Guarav et al. [13] was taken as reference, which compared the diameters of the maxillary canals (mean 1.05 SD ± 0.32) and mandibular (mean 1.842 SD ± 0.294), resulting in a sample size of at least 80 molars.

### 2.2. Collection, Storage of Teeth and Formation of Groups

It must be highlighted that: only the teeth of children whose parents had signed the donation consent were collected; the extraction was indicated according to the dentists’ treatment plan for each patient without having any relationship with this study; lastly, neither the patient nor the dentist received any financial compensation for the tooth donation.

Once the teeth were collected, they were sanitized with a 4% soapy chlorhexidine antiseptic gel to remove any organic tissue.

Subsequently, they were disinfected by immersing them in 3% [5] sodium hypochlorite for a week, to finally store in a 9% sodium chloride solution at a temperature of 4 °C. After, two working groups were established: group A of deciduous maxillary molars (subgroup A.1: first molars, subgroup A.2: second molars); group B of mandibular molars (subgroup B.1: first molars, subgroup B.2: second molars).

### 2.3. Image Acquisition and 3D Reconstruction

The teeth were mounted on heavy silicone blocks of the coltoflax^®^ brand, for their durability during subsequent handling. The images were obtained using CS 8100 ^®^ tomographic equipment (Carestream Dental, Atlanta, GA, USA) and technical specifications of 90 kVp: 15 mA and exposure of 0.75 μm. For the analysis of the images, the CS 3D Imaging and 3D Slider version 4.11.202110226 programs were used [21]. This allowed the three-dimensional reconstruction of the roots and root canals (Figure 2); additionally, they were able to carry out the measurements of the volumes (mm^3^) and surfaces (mm^2^).

### 2.4. Evaluation of Canal Morphology

The models obtained by three-dimensional reconstruction were analyzed, the roots were counted and the canals were typified, according to the Weine’s classification (Figure 3) [20].

### 2.5. Root and Canal Length Measurement

To standardize the portions to be studied, the beginning of the root cone was established by drawing a horizontal line that crossed the root furcation, and then the reconstructions were segmented (Figure 4a). Using the “line” tool in the 3D Slicer software, two lines were drawn and measured from the uppermost portion of the segmented portion to the root apex (root length) and another to the apical foramen (canal length).

### 2.6. Root Canal Angulation

Using the “angle” tool in CS 3D Slider software Carestream (Rochester, NY, USA). An angle was drawn and measured; where one of its sides followed inside the center of the longitudinal axis of the root canal; the vertex was positioned at the most outstanding point of the initial curvature; the second side ran along inside the center of the remaining root canal portion to the apical foramen. The supplementary angle was written down to maintain the parameters indicated by Schneider et al. (Figure 4c) [22].

### 2.7. Root Canal Diameter Measurements

In the three-dimensional reconstruction, the greatest length was measured in the mesiodistal (MD) and buccolingual/palatal (BL/P) directions at the level of the coronal (C), middle (M), and apical (A) thirds (Figure 4b). This measurement was corroborated in the axial CBCT images (Figure 5).

### 2.8. Volumes and Surfaces Measurement

The anatomical reconstructions were made, using the Hounsfield scale that measures the air and dental structure. Using the “quantification” tool in the 3d-Slicer software the volume and surface for each segment were calculated and reconstructed.

### 2.9. Calibration, Pilot Test, Internal Validity

The measurements were completed by a pediatric dentist specialist, who was the researcher in charge of the data collection and was trained by a specialist professor in maxillofacial radiology, for the CBCT observation. In the first phase, in the interpretation of the CBCT images, and the second phase, in the use of the three-dimensional reconstruction software, the inter-observer concordance was evaluated using the kappa coefficient 0.98 (95%CI 0.97–0.99).

A pilot test was carried out with 20 teeth to verify the feasibility, standardization, and reproducibility of the data collection process. After 15 days, the whole measurement process was carried out again, from the three-dimensional reconstruction of the root to taking measurements, volume, surface, angulation, length, and diameters. The intra-observer concordance was assessed through the interclass correlation coefficient (ICC). The ICC results obtained for volume were 0.97 (95%CI 0.96–0.99); surface 0.97 (95%CI 0.96–0.99); angulation 0.97 (95%CI 0.95–0.98); length 0.98 (95%CI 0.96–0.99); and diameters 0.95 (95%CI 0.91–0.97).

### 2.10. Statistical Analysis

The IBM SPSS vs. 26 program (Armonk, NY, USA) was extracted for the statistical study. The descriptive statistics, such as frequencies, means and standard deviation, were obtained. The significance of *p*-value < 0.05 and a power of 80% were set. Normal distribution was confirmed with the Shapiro–Wilk test. The frequencies of the qualitative variables were compared with the chi-square test, while the quantitative variables were compared with the Student’s *t*-test.

## 3. Results

### 3.1. Sample Distribution

The form of recruitment and sample selection can be seen in Figure 6.

### 3.2. General Characteristics

It was discovered that 88% of the mesiobuccal roots presented a single canal (Weine type I); the greatest morphology disadvantage was found in the lower molars, Weine’s type III was present in 59.46% of the mesial roots and in 41.66% of the distal roots (*p* < 0.05). Finally, 43.39% of the palatal roots were fused to the mesial root and there were two cases of supernumerary root (4.54%) in the lower first molars (Table 1).

In the maxillary, the palatal roots presented a length greater than 7.20 (±1.22) mm compared to the MB and DB roots. The root portions of the second molar had a mean length of 8.31 (±1.31), being longer than that of the maxillary first molar. In relation to the total axial lengths of each root of the molars studied, the data obtained are shown in Table 2.

The greatest root length reported was 11.38 mm, which corresponded to the palatal root of the upper second molar, and the minimum was 3.78 mm, corresponding to the distobuccal root of the lower first molar. The comparison of the root lengths is detailed in Figure 7.

### 3.3. Specific Characteristics

The characteristics extracted from the measurements of the root canals, such as length, angle, surface and volume, are detailed in Table 3; meanwhile, the comparisons of the means of the canals are plotted in Figure 8, where the minimum, maximum, and atypical values are also outlined.

The MB and P canals of the maxillary molar present means of 6.46 (± 1.15) and 6.39 (±1.33), being greater than the DB canals, while in the lower maxilla the buccal canals of the mesial roots presented a higher mean (7.12 ± 1.27) than the distal or mesiolingual canals. The minimum value recorded was 2.44 mm, corresponding to the DB root of the upper second molar, while the maximum value was 10.8 mm. Additionally, two lingual collateral canals of the mesial root of the maxillary second molar were found with a mean of 7.291 (±0.458). The supernumerary root canals of the lower first molars presented a mean of 6.167 (±0.975).

The angulation of the mandibular mesiobuccal root canals presented a mean of 18.74 ± 7.91. A total of 20% of the MB canals of the upper first molar presented an angulation of 40° or greater and a variance of 49.79.

The mesiobuccal and distobuccal root canals presented volumes of 3.04 (±0.13), which compared to the other root canals, were smaller and presented less variance (0.01) with a statistically significant difference of *p* < 0.05.

Regarding the diameter of the root canals studied, the collateral canals have mesiodistal diameters similar to those of the mesiobuccal canal (*p* > 0.05), with a cervical mean of 0.62 (±0.25), in the middle third of 0.33 (±0.15), and in the apical of 0.17 (±0.02). The accessory root canals in the lower molars presented similar means to the buccal canals of the distal roots (*p* > 0.05) in the mesiodistal direction, at the average level of 0.39 (±0.06) and apical 0.158 (±0.025); and in the buccolingual direction at the apical level 0.24 (±0.09). Overall, the means obtained are shown in Table 4, and the comparisons are visible in Figure 9.

## 4. Discussion

The variability of the root morphological characteristics of the primary molars represents a great challenge for pediatric dentistry. The capricious anatomical features that nature provides to the deciduous tooth are generated in part by a time-limited odontogenesis. For this reason, we emphasize that research in this regard is necessary, not only in order to strengthen the data available in the scientific literature, but also to enrich current knowledge on this matter. In this sense, this study could provide relevant information to improve, if possible, the methodology of pulp therapy.

To date, most investigations have only focused on the posterior dental group [1,2,5,6,7,8,9,11,12,13,14,15,16,17,19], as this is the most anatomically complex and further compromises the complete removal of the radicular pulp. The relevance that researchers give to the first deciduous molars is striking, as six studies focused only on that tooth [5,6,7,8,11,19]. The largest sample size studied was 487 dental units, in this case, in an in vivo study [16]. Only one in vitro study surpassed ours in this aspect [8]. In most of them, the number of molars analyzed was less than 132 dental units [1,2,5,6,7,9,11,13,14,15,23] oscillating the range between 20–90 teeth.

### 4.1. General External Characteristics of the Root Morphology of the Maxillary Molars

The number of roots present in the maxillary molars varies from two to four [2]. The three-root variant was the most common in our research, regardless of whether it was a first or second molar. The maxillary molars, which have four roots, were not found. However, the modality with two roots was presented by 54 (19%), 55 (9%), 64 (8%), and 65 (8%). Based on this result, it can be affirmed that this peculiarity was more frequent in the first molar and on the right side. Similarly to this study, Datta et al. and Wang et al. [9,14] found two roots in both of the maxillary molars studied. This fact, according to some researchers, is exceptional since, based on the results obtained, 100% of all maxillary molars have three roots [1]. The double root arrangement is a consequence of the fusion of the root portion, DB and P, which, according to some studies, is usually more frequent in the maxillary first molars [3,4,14,15]. This statement was shared based on the results of the present study.

The root fusion at 54 and 64 is sometimes quite frequent, reaching 77.7% of the total [1]. Although root fusion is not common in second molars, Bagherian et al. found in four of 14 teeth studied, that the DB and P roots were fused [1]. However, some researchers observed that on both of the maxillary molars, most of the DB and P roots were fused; this fact coincides with our results, since 46.9% of the first molars presented a fusion, and 39.1% of the second molars [8].

In evaluating each root cone length of all of the maxillary molars, Gaurav et al. [13] determined that the mean in the P root (8.03 mm) represented a greater length with regard to the MB (7.75 mm), and the DB root was the shortest (7.61 mm). However, according to the data from our results, the longest was the MB (7.60 mm) followed by the palatine (7.20 mm), and in the same way, the DB was the shortest (6.74 mm). Therefore, we are available to affirm that globally the dimensions of the roots in this study are smaller. When studying each molar individually, there was no consensus among the different authors. According to Bagherian et al. [1], in the first molars, the longest root was the MB with a mean of 8.11, followed in size by the P with a length of 7.14 mm, and the DB was the one with the shortest length, with a mean of 6.77. These authors share with Fumes et al. that the mean length of the MB root was the largest (7.9 mm), however, they differ from the other two pieces of research, since P was the one with the shortest length (5.9 mm) compared to root DB (6.7 mm) [7]. In contrast, Zoremchhingi et al. reflected in their research that the distobuccal roots were the longest, with an average length of 7.32, the palatal the shortest (6.72 mm), and the mesiobuccal the second-longest (6.88 mm) [15]. The first molars were evaluated to represent a smaller longitudinal dimension in the three roots compared to other studies, nevertheless, this study agreed with most of the research, that the largest root was the MB (6.58 mm), followed by P (5.98 mm), and finally DB (5.57 mm). This same sequence was observed in the second molars: MB (8.61 mm); P (8.43 mm); and DB (7.90 mm), clearly exceeding the dimensions of the first molars. Based on the data from the other studies, the sequence was different since the root P was the largest, representing an average length of 9.92 mm and the MB root was the second-longest (between 8.24 mm and 9.57 mm) [1,15]. Other investigations have observed, as this research did, that the longest was the MB, reaching an average of 8.5 mm, a result similar to that of the molars in our study [7]. We agree, on this occasion, with all of the authors, that the shortest was the DB root (8.06 mm, 7.90 mm, 7.21 mm, and 6.5 mm, respectively) [1,7,15]

### 4.2. General Internal Characteristics of the Root Morphology of the Maxillary Molars

All of the roots of the maxillary first molar present only one canal in a high percentage (MB = 89.66% to 93.3%), (DB = 95.65% to 100%), (P = 100%) [1,5,8,13,14,15]. However, in this study, the results described a lower percentage with respect to the DB canal (53.5%), since the MB (100%) and P (93.8%) are similar. The prevalence of a second canal in the mesiobuccal roots is variable between the maxillary molars according to research (6.67% to 95%) [3,4], and, when they occur, their presence is more frequent in the maxillary second molars [14]. The results of this study agree on this aspect. Because they do not exist in the maxillary first molar and the maxillary second molar, they are only present in 13%. In contrast to other investigations in 55 and 65, a canal was found for each root. (MB = 46.6% to 100%), (DB = 73.3% to 100%), and (P = 60–100%). The authors of some studies [1,13,15] documented two root canals in 53.3% of the samples; according to other authors, up to three mesiobuccal canals [4,15] and two complex configurations were found between the DB and P root canals [4]. When these roots merge, the number of canals is usually two [14]. The DB root has a single canal in all teeth [4], however, two distobuccal canals have been found in canals in the second maxillary molars, as identified by Zoremchhingi et al. [15], in whose samples, 26.6% had two root canals.

### 4.3. General External Characteristics of the Root Morphology of the Mandibular Molars

Mandibular primary molars can have from one [24] to three roots; in the latter case, according to Moyano et al., the prevalence was 0.44% for the first molars and 0.22% for the second molars, with the predisposition for this being higher amongst males and in the right side of the arch [17]. The mesial double root variant is the most common [3,4] and frequent variant [1]. In this regard, a single M root with apical bifurcation was observed, but in general, the first molars have presented two roots [14,15]. Although cases are described in the literature of a mandibular first molar with four roots, this is exceptional [25].

The finding of two supernumerary roots in the lingual face of the two first mandibular molars represents a great curiosity (Figure 10a). Typically, the second molars have two to three roots [14,15]. A total of 95.5% and 72.28% of these molars have two roots, and only 1 of 22 molars studied, or 27.52% according to other researchers, had three roots [1,16]. There are differences between genders in the prevalence of an additional root and the incidence of symmetry [16]. Accessory roots were observed in second molars among Asian population groups [3]. Note that the origins of our teeth were European.

Regarding the average length of the mandibular roots of both of the lower molars, we agree with the scientific literature on the sequence, reflecting that the M root was the longest with an average length of 8.28 mm, while the D root measured 7.18 mm [13]. Again, our means represent lower dimensions.

According to the different investigations, in the mandibular first molar, the mesial roots have an average length of 9.66 mm and 9.45 mm, and we shared the sequence of the distal ones in our study of 7.22 mm and 8.42 mm, but once again our dimensions were smaller (7.58 mm and 6.57 mm). The same characteristics represented the second molars, obtaining a mean length of 7.89 mm in the mesial roots and 7.62 mm in the distal ones, compared to the mean lengths of other researchers: 9.40 mm and 10.67 mm in the mesial roots; in the distal ones 8.27 mm and 9.83 mm [1,9].

### 4.4. General Internal Characteristics of the Root Morphology of the Mandibular Molars

According to Bagherian et al., 81% of mandibular first molars have two mesial canals and 22% have two distal canals in our research; the frequency with which the two mesial and distal canals were present was 51.1% and 38.2%, respectively. The percentages found by Katge et al. were also higher when finding two mesial canals in 80% of the first molars, in contrast, the percentage of the two distal ones was lower (23.3%). In the second molars, 100% presented two mesial canals and only 36.4% presented two distal canals. However, our percentages were lower concerning the mesial canals (73.5%) and higher concerning the distal canals (44.1%). Katge et al. observed in the second molars two mesial canals in 100% of the cases, and in 56.6% of the cases, two distal canals [5]. Datta et al. [9] observed four canals in 81.2% of the first molars and 87.5% of the second molars, percentages lower than those in our study. In contrast to these results, what is usual according to other researchers is that the mandibular group presents one M root canal and two distal root canals [13]. To Katge et al., 80% of the first molars have two mesial canals and 23.3% of the first molars have two distal canals. When studying the mandibular second molars, 100% have two mesial canals and 56.6% have two distal canals [5].

Lateral canals have been observed in the mesial and distal roots of the mandibular second molars, some of them emptying into the cementum and others into the dentine [4]. The root canal configurations vary with age, particularly in the mandibular molars.

### 4.5. Root Canal Characteristics of the Maxillary and Mandibular Molars

#### 4.5.1. Typology of Root Canals

The Weine’s classification is the most widely used to describe the morphology of root canals in deciduous teeth, since it does not underestimate or overestimate the presence of root canals in CBCT. It is also easy to calibrate for the clinician, although it considers the root and not the tooth [1,20]. This classification was used because of all of its advantages and the design of this study. However, other classifications are more detailed, such as Vertucci and Ammed. Vertucci has greater utility in other experimental designs that use a clearing and ink staining technique, that helps to visualize the root canals’ branching that cannot be seen with CBCT [5]. On the other hand, Ammed presents as an advantage the description of the canals based on the analyzed tooth and not on the dental root; however, a great disadvantage is that the resulting codes are very long, and it also considers root canals that require a micro-CBCT design to be analyzed [4].

At the level of the maxillary molars and considering the Weine’s classification, the research reflects that the maxillary first molars that most frequently have a MB canal are type I (92.6% or 93.10%); therefore, it coincides in this aspect, and in our study the percentage for these same canals is 93.3%. Regarding the DB canal (96.3% or 96.65%) and P (100%), type I is again the most prevalent in all of the molars [1,5]. In this study, although type I P was the most frequent (81.3%), the same did not occur with BVD since it represents a type I typology in 46.4% of the molars and the same percentage of type III, that is, they are equally present in the DB canal. In the maxillary second molars, our results show how type I occurs in a higher percentage in the MB (69.6%) and P (85.7%) canals; however, in the DB root canal, the most prevalent type is type III (43.5%). Other investigations have determined that all or nearly all of the maxillary second molar canals are type I [1,5]. Regarding the mandibular molars, specifically, the first molars, a large percentage (81.5% or 73.3%) of the mesial canals are type IV, however, the distal canals are mainly type I (77.8% or 76.6%) [1,5] We disagree with these authors since the most frequent M root canal is type III (44.4%); in this study, as the D root canal is type I, we agree with the other investigations. At the level of the second molars, the root canals are type IV (100%; 100%) in the M root and type I (63.6%) in the D, although others have seen that in this canal, although it is type IV, the percentage is lower [1,4]. We agree on the typology of the mesial canal but not on the distal, which in our case was the most prevalent type III (41.2%). Apart from these results, according to other authors, most of the compounds do not fit into any classification [2].

#### 4.5.2. The Axial Length of the Root Canal

In the maxillary first molar, the longest length of the canal, as in our investigation (5.6 mm), corresponds to the MB; its measurement is 6.5 mm, followed by the DB with 5.4 mm, and the shortest is the P since it measured 4.6 mm. [7]. According to our results, the DB root canal was the smallest (4.6 mm). In the second molar, although the VB canal is the largest (6.3 mm), the smallest canal corresponds to the DB (5.7 mm), as the P canal is slightly larger (5.9 mm) [7]. In this study, the MB and P canals are similar in length (7.2 mm), however, the DB is the shortest (6.7 mm). The mean length of the root canals of the mandibular first molar is 6.1 mm in M and 4.7 mm in D [7]. In our investigation, the mean length of the mesial canals was 5.4 mm and 4.9 mm in the distal ones. In the mandibular second molar, the M measured 7 mm and the distal 6.7 mm. [7] According to our results, the values obtained were 6.3 mm and 5.9 mm, respectively. In all of the cases, the mean length of the mesial canal exceeded the distal.

#### 4.5.3. Two-Dimensional Surface in mm^2^ of the Root Canals

The results relative to the surface per mm^2^ can be compared with the study by Fumes et al. [7] There are discrepancies in this regard, the largest referring to the mandibular second molar; according to both results, for the M canal there is a two-dimensional difference of 33 mm^2^ and for the D, a difference of 30.7 mm^2^, based on the size of the canals of this molar, they are smaller in our study. The smallest discrepancy was found in the distobuccal and palatal canals of the maxillary second molar; the two-dimensional differences are 2 mm^2^ and 2.1 mm^2^, respectively. In our investigation, the MB canal is larger, however, the P is smaller.

#### 4.5.4. Three-Dimensional Volume in mm^3^ of the Root Canals

The mean root pulp volume of all of the maxillary molars according to Nevoda et al. [20] is 43.2 mm^3^ and that of the mandibular molars is 30 mm^3^. Our results reflect a smaller three-dimensional dimension, obtaining a total of 29.7 mm^3^ in the maxillary molars and 37.7 mm^3^ in the mandibular ones; These data, compared with those of Fumes et al. [7], indicate a greater dimension of the molars for these authors, obtaining 17.6 mm^3^ in the volume maxillary molars and 26.2 mm^3^ in the mandibular ones. By canals, in our research compared to Fumes et al., the greatest and least three-dimensionality corresponded to the palatine canals in both of the maxillary molars (4.71 and 6.89 mm^3^, respectively); and with the mesiolingual canals (5.06 mm^3^) in the maxillary first molars and distolingual canals (7.65 mm3) in the second molars. Concerning the lower volume observed in the maxillary molars, the DB canals in both (2.26 and 3.68 mm^3^) as well as in Fumes et al. and the distobuccal canal (3.48 mm^3^) in the mandibular first molar.

#### 4.5.5. Mesiodistal and Buccolingual Diameters of the Root Canals

Another aspect to consider in the internal root morphology study, in particular, is the choice of the most suitable endodontic files for canal instrumentation. Unlike us, the researchers determined the largest and smallest diameters at different levels but without considering both sections [15]—the mesiodistal and buccolingual—sometimes not even considering the order of the molars studied [13], and determining only the canal shape and not its diameter [14,18]; in this sense, our research is pioneering. According to our measurements, in the mesiodistal direction, the largest diameters are located in the palatine canals of both of the maxillary molars (1.17 and 1.19), a fact that we share with other researchers [13,14]. The palatine canal represents the largest diameter in the maxillary group (2.56, 1.0, and 1.3), and those of smaller dimensions in the distobuccal canals (0.18 and 0.16). In the lower arch, the largest diameters of this section correspond to the mesiobuccal canals of the mandibular first molar (0.84) and the distolingual canals of the mandibular second molar (0.8), data that do not coincide with other investigations [13], in which the diameter of the distobuccal canal is the largest in the mandibular group (2.48) and in the distal canals (1.1 and 1.6) of this same group [14]. The smallest dimensions are located in the mesiobuccal canals of both of the mandibular molars (0.16 and 0.19). In the measurements obtained in the buccal–lingual direction, there is a greater variability in their location in the different canals, in such a way that the largest diameters of this section are housed in the distobuccal canals in the maxillary first molars (2.34) and mesiobuccal canals in the second (2.33), the smaller dimensions correspond to the distobuccal canals (0.51) in the maxillary first molar and mesiobuccal canals (0.21) of the second maxillary molar. In the mandible, the largest dimensions are housed in the distobuccal canals (2.37) in the first molars and the distolingual canals (2.39) in the second molars. The smallest diameters are found in the mesiobuccal canals in both molars (0.27 and 0.33).

#### 4.5.6. Curvatures of the Root Canals

The data found in the literature regarding the angulation of the canals refer to the prevalence [5,8] of this characteristic. In the maxillary first molars, the most frequently presenting this angulation is the MB canal (67%; 72.41%) and in the maxillary second molars the palatine canal (88.89%) or MB (78%), according to the different researchers. Regarding the mandibular teeth, the most prevalent curved canal is the MB (70.8%) or M (50%) in the first molars, and the D (61.5%) or M (55%) when it is a single canal in the second molars. However, in our research, the mean angulation of all of the root canals studied was determined, with the MB canal being the one with the greatest curvature (21.2°) in the maxillary first molars and the DB in the second molars (24.3°). Regarding the mandibular molars, in the first, the canal with the greatest angulation is the mesiobuccal (25.9°), and the distobuccal (21.9°) in the second mandibular molars. Other authors choose to record the root angulation [1,13,15], whereas we believe that it is more relevant to determine the angulation of the canals concerning possible pulp treatment. We believe that, in common with other researchers, considering the angulated morphology of the canals in the apical third, a careful selection of mechanical or rotary instruments and additional disinfection, such as passive ultrasonic irrigation or negative apical pressure is advised [7], which must be very careful to avoid dentine damage that could compromise the treatment.

In addition to all of these morphological and dimensional aspects, we must consider the presence of the horizontal anastomoses, lateral canals, and wide buccolingual root canals [11]. We believe that it would be interesting to continue investigating the root morphological study of primary molars in search of unusual forms and an improvement in the knowledge and application of therapeutics that are already described in the literature descriptions of unusual canals; for example, one of them refers to the mandibular canals of the mandibular first molars, observing that sometimes they can be C-shaped, this anomaly extending from the cervical third to the apex and categorized as Melton type I, which would severely complicate pulp treatment [18].

A study limitation was the lack of donor’s descriptive data (sex and age) because the teeth were extracted by dentists who were unconnected to this study and were not indicated to them to write down the sex and age of each tooth collected, and for this reason, the exact data on the individual sources of each tooth is unknown, although the mixed dentition can be indicated as the stage dentition that all donors had. However, in the literature, only one study has been found [16] that indicates the number, sex, and age of the donors who participated in the study, but these data were not used to classify the sample.

Finally, we believe that CBTC applied to the morphological study of the primary dentition, in addition to optimizing knowledge about the root pulp, could be a matter of interest to promote not only the impact on research, but also beyond the limits of the academic environment [26].

## 5. Conclusions

The root morphology of deciduous molars presents great variability, especially at the level of their root canals. The findings of this research will allow pediatric dentists to know the variations in the external and internal morphological characteristics of the root portion of maxillary and mandibular deciduous molars in order to be able to anticipate possible complications derived from inadequate pulp techniques. Two supernumerary roots were found in the mandibular first molars in a lingual position. All of the roots of the maxillary first molar present only one canal was found in a high percentage of the sample, and the frequency with which the two mesial and distal canals are presented is 51.1% and 38.2%, respectively. Weine’s type I is the most frequent in all of the molars. The mesiobuccal canal of the second molar is the one that presents a considerable angulation with greater frequency. In the mandible, the largest dimensions are seen in the distobuccal canals and the most prevalent curved canal in the mesiobuccal root canal.

## Figures and Tables

**Figure 1 ijerph-19-09162-f001:**
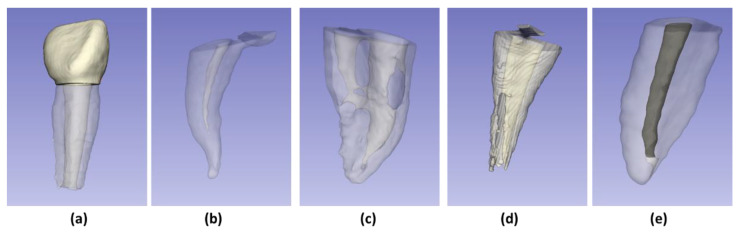
Examples of excluded samples: (**a**) Anterior tooth; (**b**) Root canal obliteration; (**c**) Perforating root resorption; (**d**) External root resorption; (**e**) Root canal with pulp therapy.

**Figure 2 ijerph-19-09162-f002:**
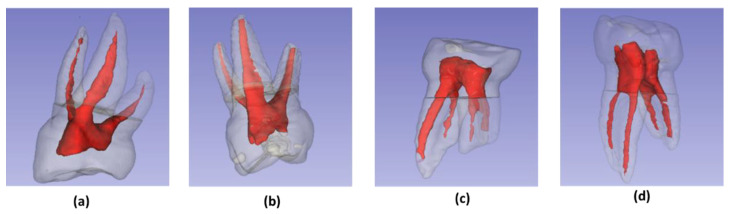
Three-dimensional reconstructions: (**a**) Maxillary 1st molar; (**b**) Maxillary 2nd molar; (**c**) Mandibular 1st molar; (**d**) Mandibular 2nd molar.

**Figure 3 ijerph-19-09162-f003:**
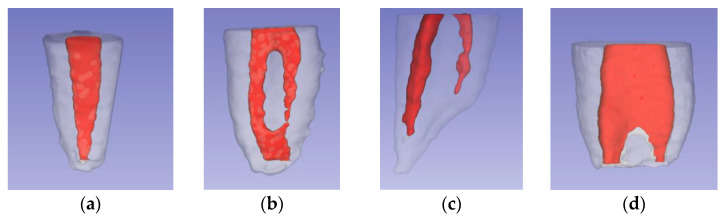
Weine’s classification of root canal [20]: (**a**) Type I: single canal and apex; (**b**) Type II: two canals fused at the apical level; (**c**) Type III: two canals and two apexes; (**d**) Type IV: single canal that divides and ends in two apexes.

**Figure 4 ijerph-19-09162-f004:**
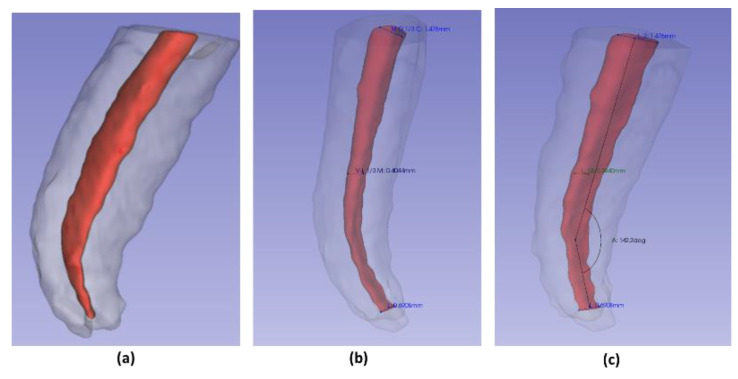
Segmentation and measurements: (**a**) root segmentation; (**b**) diameter measurement; (**c**) angle measurement.

**Figure 5 ijerph-19-09162-f005:**
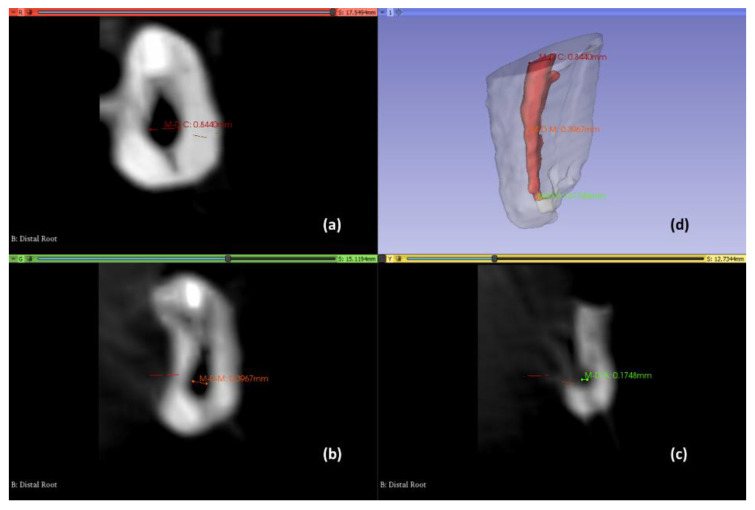
Axial views to determine the root canal diameter and three-dimensional reconstruction. Coronal third (**a**); Middle third (**b**); Apical third (**c**); three-dimensional reconstruction view (**d**).

**Figure 6 ijerph-19-09162-f006:**
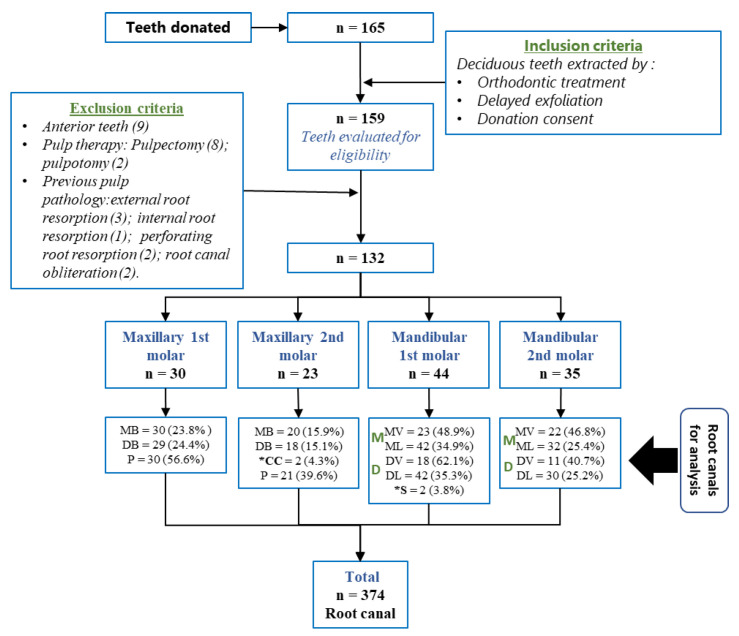
Recruitment and sample distribution. MB (mesiobuccal canal); ML (mesiolingual canal); DB (distobuccal canal); DL (distolingual canal); *S (supernumerary root canal), *CC (collateral canal).

**Figure 7 ijerph-19-09162-f007:**
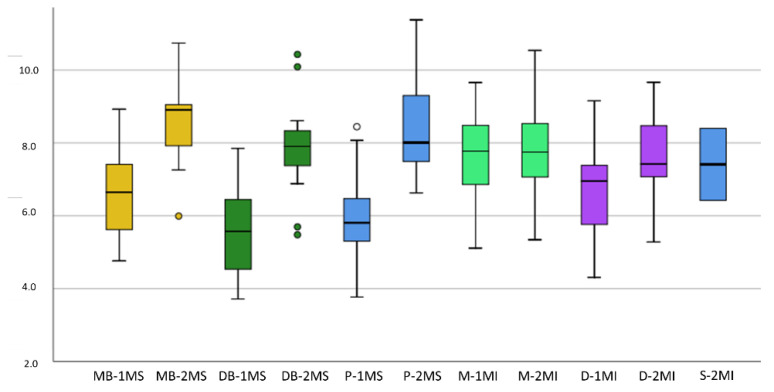
Box plot for comparison of root length: Axis Y: root length (mm), Axis X: 1MS (Maxillary 1st molar), 2MS (Maxillary 2nd molar), 1MI (Mandibular 1st molar), 2MI (Mandibular 2nd molar), MB (mesiobuccal), DB (distobuccal), P (palatal root), M (mesial), D (distal), S (supernumerary).

**Figure 8 ijerph-19-09162-f008:**
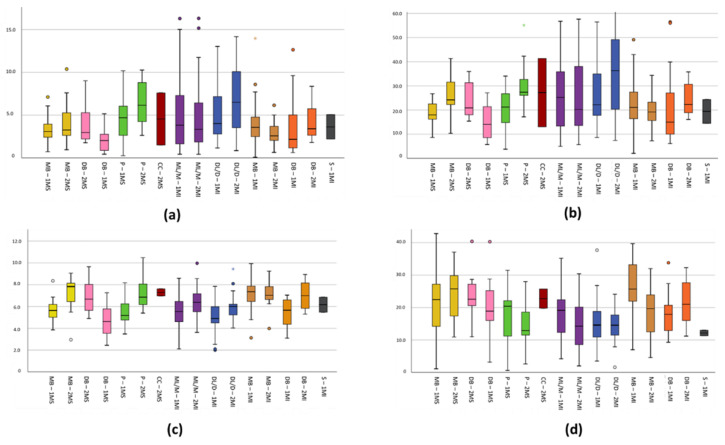
Box plot for comparison of root canal variables: In axis Y outlined (**a**) Length (mm), (**b**) Angle (°), (**c**) Surface (mm^2^), (**d**) Volume (mm^3^). Axis X is described as root canals 1MS (Maxillary 1st molar), 2MS (Maxillary 2nd molar), 1MI (Mandibular 1st molar), 2MI (Mandibular 2nd molar), MB (mesiobuccal), DB (distobuccal), P (palatal root), ML/M (mesial), DL/D (distal), S (supernumerary).

**Figure 9 ijerph-19-09162-f009:**
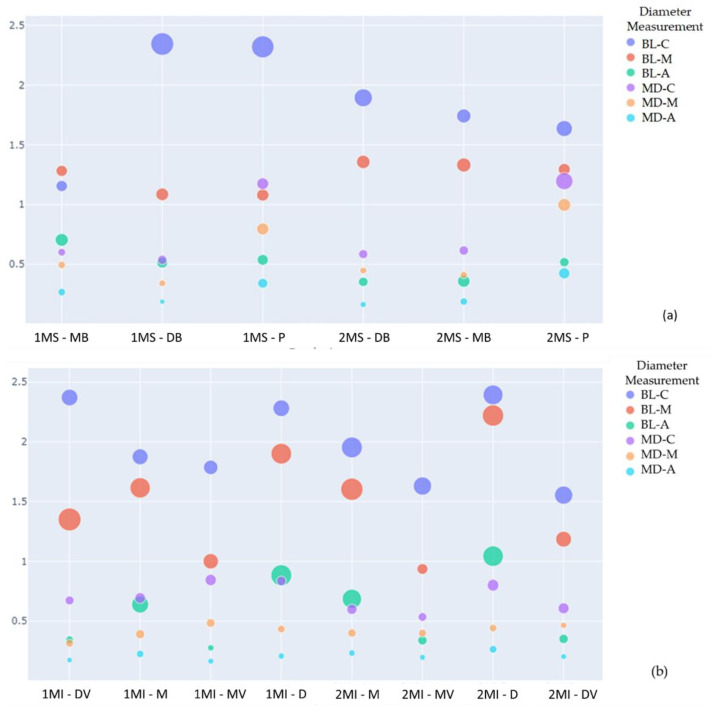
Scatter plot for comparison of each canal diameter. Where it is possible to observe the mean diameters of the canals in the buccolingual (BL) and mesiodistal (MD) directions at the three apical (A), middle (M), and cervical (C) levels of the canals of the upper molars. The size of the circumference indicates the increase in the standard deviation (SD). Axis Y: Diameter measurement (mm). Axis X: 1MS (Maxillary 1st molar); 2MS (Maxillary 2nd molar); 1MI (Mandibular 1st molar); 2MI (Mandibular 2nd molar); MB (mesiobuccal); DB (distobuccal); P (palatal root); M (mesial); D (distal); S (supernumerary); (**a**) Maxillary molar comparison; (**b**) Mandibular molar comparison.

**Figure 10 ijerph-19-09162-f010:**
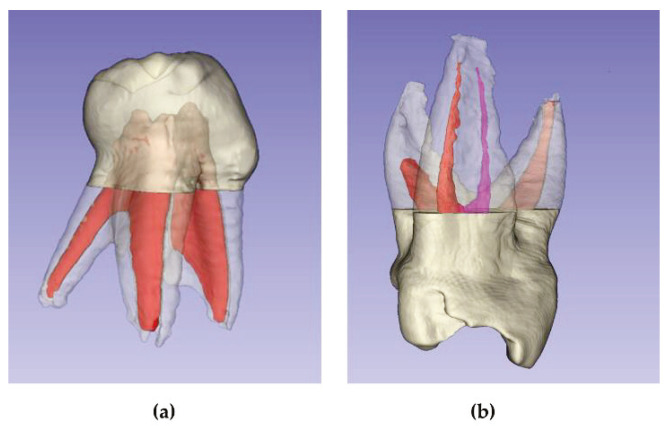
Anatomical peculiarities: (**a**) Mandibular 1st molar with supernumerary root; (**b**) Collateral canal in mesial root.

**Table 1 ijerph-19-09162-t001:** Root morphology distribution of deciduous molars.

		Maxillary			Mandibular	
Weine’s Classification		1M	2M		1M	2M
	Root	Frequency (%)	Frequency (%)	Root	Frequency (%)	Frequency (%)
I	MB	28 (93.3%) *	16 (69.6%) *	M	14 (31.1%)	7 (20.6%)
II			2 (8.7%)		3 (6.7%)	1 (2.9%)
III			1 (4.3%)		20 (44.4%) *	24 (70.6%) *
IV		2 (6.7%)	4 (17.4%)		8 (17.8%)	2 (5.9%)
I	DB	13 (46.4%)	9 (39.1%)	D	18 (40%)	13 (38.2%)
II			2 (8.7%)		1 (2.2%)	1 (2.9%)
III		13 (46.4%)	10 (43.5%)		16 (36.6%)	14 (41.2%)
IV		2 (7.1%)	2 (8.7%)		10 (22.2%)	6 (17,6%)
	Fusion PR + DBR	14 (46.6%)	9 (39.1%)			
I	P	13 (81.3%)	12 (85.7%)	S	2 (100%)	
II			2 (14.3%)			
III		1 (6.3%)				
IV		2 (12.5%)				

* Chi-square test, statistical significance at *p* ≤ 0.05. 1M (first Molar); 2M (second Molar); M (mesial); D (distal); P (palatal); MB (mesiobuccal); DB (distobuccal); PR (palatal root); DBR (distobuccal root); S (supernumerary root).

**Table 2 ijerph-19-09162-t002:** Root length.

**Root**	**Maxillary**			
	**1M**		**2M**	
	**Mean**	**SD**	**Mean**	**SD**
MB	6.58	1.14	8.61 *	1.15
DB	5.57	1.23	7.90 *	1.41
P	5.98	1.07	8.43 *	1.38
	**Mandibular**			
	**1M**		**2M**	
	**Mean**	**SD**	**Mean**	**SD**
M	7.58	1.27	7.89	1.35
D	6.57	1.20	7.62 *	1.21
S	7.41	1.39		

Student’s *t*-test, * statistical significance at *p* ≤ 0.05. SD: Std. Deviation. 1M (1st molar); 2M (2nd molar); M (mesial); D (distal); S (supernumerary root); P (palatal); MB (mesiobuccal); DB (distobuccal).

**Table 3 ijerph-19-09162-t003:** Specific characteristics of root canals.

Tooth	Root Canal		Length (mm)		Angulation (Degree)		Surface (mm^2^)		Volume (mm^3^)	
			Mean	SD	Mean	SD	Mean	SD	Mean	SD
Maxillary										
1M	MB		5.68	0.88	21.2	11.68	18.66	4.79	3.42	1.53
	DB		4.66	1.30	20.32	8.46	15.54	7.46	2.26	1.50
	P		5.58	1.26	17.58	8.30	21.06	8.03	4.71	2.66
2M	MB		7.25 *	1.42	23.83	7.41	26.28 *	7.40	4.21	2.38
		CC	7.29	0.45	22.75	4.17	27.19	19.99	4.55	4.30
	DB		6.75 *	1.39	24.03	7.28	24.20 *	7.56	3.68 *	1.89
	P		7.20 *	1.39	15.25	6.04	29.70 *	9.22	6.89 *	3.92
Mandibular										
1M	M	B	7.06	1.41	25.98 *	8.21	23.98	11.42	4.34	2.96
		L	5.49	1.39	18.72 *	7.46	26.09	14.39	5.06	4.14
	D	B	5.45	1.27	18.00	6.43	21.81	15.71	3.48	3.35
		L	4.98	1.31	14.87	6.77	26.79	12.63	5.02	3.19
	S		6.16	0.97	12.15	1.20	19.42	6.97	3.63	2.01
2M	M	B	7.18	1.13	18.74	7.61	19.67	6.10	2.91	1.36
		L	6.31	1.38	14.67	7.81	25.71	16.84	5.10	5.11
	D	B	7.05 *	1.34	21.95 *	7.33	24.85	7.01	4.22	2.07
		L	5.94 *	1.16	14.45	4.97	35.28 *	15.79	7.65 *	5.71

Student’s *t*-test, * statistical significance at *p* ≤ 0.05. SD: Std. Deviation. 1M (first molar); 2M (second molar); MB (mesiobuccal); DB (distobuccal); P (palatal); M (mesial); D (distal); S (supernumerary); CC: collateral canal; S: supernumerary root canal; B (buccal); L (lingual).

**Table 4 ijerph-19-09162-t004:** Diameter measurement molars.

Maxillary							Mandibular							
			1M		2M						1M		2M	
Root canal	measure		Mean	SD	Mean	SD	Root	Canal	measure		Mean	SD	Mean	SD
MB	BL	C	1.15	0.34	1.68 *	0.5	M	B	BL	C	1.78	0.5	1.63	0.81
		M	1.28	0.34	1.31	0.48				M	1	0.57	0.93	0.28
		A	0.70 *	0.46	0.37	0.43				A	0.27	0.09	0.33	0.21
	MD	C	0.6	0.16	0.61	0.25			MD	C	0.84	0.3	0.53	0.19
		M	0.49	0.16	0.41	0.14				M	0.48	0.19	0.4	0.15
		A	0.26	0.16	0.18	0.16				A	0.16	0.08	0.19	0.08
DB	BL	C	2.34	1.42	1.89	0.89		L	BL	C	1.83	0.56	1.95	1.06
		M	1.08	0.44	1.35	0.49				M	1.64	1.02	1.6	1.2
		A	0.51	0.32	0.35	0.25				A	0.65	0.71	0.68	0.91
	MD	C	0.53	0.24	0.58	0.23			M-D	C	0.69 *	0.28	0.59	0.26
		M	0.34	0.13	0.44 *	0.12				M	0.39	0.21	0.4	0.17
		A	0.18	0.07	0.16	0.1				A	0.22	0.14	0.23	0.1
P	BL	C	2.32	1.37	1.63	0.71	D	B	L	C	2.37 *	0.67	1.55	0.8
		M	1.08	0.4	1.29	0.39				M	1.35	1.27	1.18	0.61
		A	0.53	0.32	0.51	0.24				A	0.34	0.15	0.35	0.21
	MD	C	1.17	0.38	1.19	0.81			MD	C	0.67	0.19	0.6	0.29
		M	0.79	0.4	0.99	0.45				M	0.31	0.15	0.46	0.09
		A	0.34	0.27	0.42	0.34				A	0.17	0.07	0.2	0.08
								L	BL	C	2.28	0.67	2.39	0.96
										M	1.9	1.04	2.22	1.11
										A	0.88	1.07	1.04	1.02
									MD	C	0.83	0.21	0.8	0.32
										M	0.43	0.14	0.44	0.14
										A	0.2	0.09	0.26 *	0.14

Student’s *t*-test, * statistical significance at *p* ≤ 0.05. SD: Std. Deviation. 1M (first molar); 2M (second molar); MB (mesiobuccal); DB (distobuccal); P (palatal); M (mesial); D (distal); B (buccal); L (lingual); BL (buccal-lingual direction); MD (mesiodistal direction); C (cervical third); M (Middle third); A (apical third).

## Data Availability

The datasets generated and analyzed during the current study are not publicly available for privacy reasons, but they are available from the corresponding author on reasonable request.

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
