# Peer review of "Cone-Beam Computed Tomography (CBTC) Applied to the Study of Root Morphological Characteristics of Deciduous Teeth: An In Vitro Study"

_ijerph, 2022, doi:10.3390/ijerph19159162_

Round 1

Reviewer 1 Report

In general the article is valid, but some points listed below need to be improved before publication.

Introduction:

There are too many bibliographic references for any single sentence. They need to be reorganized.

Methods: 

Explain how the measurement took place? How many operators have performed it? Has a calibration been done?

Results:

Specify in more detail the inclusion and exclusion criteria also in the materials and methods section.

Discussion:

Describe the limitations of the study at the end of the discussion.

Conclusions: 

At the end of the conclusion, suggest practical-clinical ideas and/or related future research.

Author Response

Dear Reviewer:

Both authors appreciate your comments and suggestions, these have helped improve the way we present our research. Below we will detail and resolve each of your comments.

Reviewer 2 Report

Dear Authors I recommend minor revision. Despite I found it quite well conducted and well reported, i suggest to add some statistical description on personal data of the sample. Number of patients, sex, age etc. Moreover could be interesting to define the term "without resorption".  Please define also if the permanent teeth was present in the maxillaries in all cases considered in the sample. A more detailed description of the clinical sample feature could help the reader to better understand the teeth sample studied. Please add also a table of the subgroup numerosity divided for sex and group age.  

Author Response

(The authors gave the same response as above.)

Reviewer 3 Report

Dear Authors,

I have read the article "Cone-beam Computed Tomography (CBTC) applied to the study of root morphological characteristics of deciduous teeth: an in vitro study" which seems to be quite interesting. Eventhough I found some issues, which I would like to point and ask you for explanation.

1. within the introdction you are pointing out that "deciduous teeth root canals morphology is chalenging for profesionals". Please can you explained it more? To be honest I cannot see the point why - like if you are going to do RCT on deciduous tooth? Because for pulpotomi procedures it is not so important. Or you want to point that propper RCT is almost not possible?

2. venthough it doesn´t seem important, please provide us some better description of donors group (males vs females, average age, etc)

3. did the legal representatives give you informed consent?

4. in the chapter 2.1 you are writing that "molars without resorption" and in the inclusion criteria (fig.5) you are mentioning that you included the teeth with present at least 2/3 of the root

5. if you have teeth with "only" 2/3 of the roots you are not able to properrly distinguish Weines classification , because most of the diffences is localised within the apical 1/3 lenght

6. if only one root was resorbed, did you still use the tooth for evaluate the other potentialy non resorbed roots? I believe so, but please state it within the methodology

7. please provide some three dimensional reconstruction on which you will be able to better explain the exclusion criteria

Eventhough I cannot see the potential clinical usage of the study, the authors are presenting very nice anatomical/morphological description of the deciduous teeth root system, which is often neglected. 

Author Response

(The authors gave the same response as above.)

Reviewer 4 Report

The authors of the submitted manuscript intend to perform an in vitro study using cone-beam computed tomography (CBTC) to study the root morphological characteristics of deciduous teeth.

The following points are suggested for further consideration.

1. For the materials and methods section. Please specify the persons who are recruited to perform all the measurements; so, it is necessary to give the intraobservers and interobservers reliability.

2. The quality of radiographic pictures in figure 4 appears not clear enough. Can the authors modify them?

Author Response

(The authors gave the same response as above.)

Round 2

Reviewer 3 Report

Dear authors,

thank you for explaining or incorporating my comments into the manuscript. I believe, that it will be valuable source of information for other dentist.

Author Response

(The authors gave the same response as above.)
